# MACCA: Offline Multi-agent Reinforcement Learning with Causal Credit Assignment

**Ziyan Wang**                                                                    *ziyan.wang@kcl.ac.uk*
*King's College London*

**Yali Du**                                                                        *yali.du@kcl.ac.uk*
*King's College London*

**Yudi Zhang**                                                                     *y.zhang5@tue.nl*
*Eindhoven University of Technology*

**Meng Fang**                                                                      *meng.fang@liverpool.ac.uk*
*University of Liverpool*

**Biwei Huang**                                                                    *bih007@ucsd.edu*
*University of California San Diego*

**Reviewed on OpenReview:** *https://openreview.net/forum?id=gwUOzI4DuV*

## Abstract

Offline Multi-agent Reinforcement Learning (MARL) is valuable in scenarios where online interaction is impractical or risky. While independent learning in MARL offers flexibility and scalability, accurately assigning credit to individual agents in offline settings poses challenges because interactions with an environment are prohibited. In this paper, we propose a new framework, namely **M**ulti-**A**gent **C**ausal **C**redit **A**ssignment (**MACCA**), to address credit assignment in the offline MARL setting. Our approach, MACCA, characterizing the generative process as a Dynamic Bayesian Network, captures relationships between environmental variables, states, actions, and rewards. Estimating this model on offline data, MACCA can learn each agent's contribution by analyzing the causal relationship of their individual rewards, ensuring accurate and interpretable credit assignment. Additionally, the modularity of our approach allows it to integrate with various offline MARL methods seamlessly. Theoretically, we proved that under the setting of the offline dataset, the underlying causal structure and the function for generating the individual rewards of agents are identifiable, which laid the foundation for the correctness of our modeling. In our experiments, we demonstrate that MACCA not only outperforms state-of-the-art methods but also enhances performance when integrated with other backbones.

## 1 Introduction

Offline Reinforcement learning (RL) has gained significant popularity in recent years. It can be particularly valuable in situations where online interaction is impractical or infeasible, such as the high cost of data collection or the potential danger involved (Levine et al., 2020). In the multi-agent setting, offline multi-agent reinforcement learning (MARL) has identified and addressed some of the challenges inherited from offline single-agent RL, such as distributional shift and partial observability (Du et al., 2023). For example, ICQ (Yang et al., 2021) focuses on the vulnerability of multi-agent systems to extrapolation errors, and CQL (Kumar et al., 2020) aims to mitigate overestimation in Q-values, which can lead to suboptimal policy learning. The independent learning paradigm in MARL is appealing due to its flexibility and scalability, making it a promising approach to solving complex problems in dynamic environments. While independent learning in MARL has its merits, it will significantly hinder algorithm efficiency when the offline dataset only includes team rewards. This presents a credit assignment problem, aiming to assign credit to individual

agents within partial observability and emergent behavior. While plain reward regression methods directly map joint state-action pairs to a single scalar reward, they inherently lack interpretability, as they do not reveal which specific dimensions of states or actions drive each agent's individual contributions. In contrast, employing a causal model explicitly captures these critical dependencies, providing clarity on each agent's causal influence and enabling more effective and interpretable credit assignment.

In offline MARL, addressing the issue of credit assignment is challenging. Agents are reliant on static, pre-collected datasets, often spanning a variety of behavior policies and actions across different time periods. This diversity in data distributions increases the difficulty of assigning credits, given that the nuances of agent contributions are lost in the plethora of policies. Recent credit assignment methods, such as SQDDPG (Wang et al., 2020) and SHAQ (Wang et al., 2022a), are primarily conceived for online scenarios where continuous feedback aids in refining credit assignments. However, when restricted to static offline data in offline MARL, they miss out on the essential dynamism and agility needed to accurately understand the intricate interplay within the dataset. Moreover, in offline settings, methods like SHAQ, which rely on the Shapley value, and SQDDPG, which employs a Shapley-like approach for individual Q-value estimation, face inherent challenges. Computing the Shapley value or its approximations demands consideration of every potential agent coalition, a process that is computationally intensive. In offline MARL, such approximations can lead to imprecise credit assignments due to a loss in precision, potential data inconsistencies from the static nature of past interactions, and scalability issues, especially when numerous agents operate in intricate environments.

In this paper, we propose a new framework, namely **M**ulti-**A**gent **C**ausal **C**redit **A**ssignment (**MACCA**), to address credit assignment in an offline MARL setting. MACCA equates the importance of the credit assignment and how the agent makes the contribution by causal modeling. MACCA first models the generation of individual rewards and team reward from the causal perspective, and construct a graphical representation, as shown in Figure 1, over the involved environment variables, including all the dimensions of states and actions of all agents, the individual rewards and the team rewards. Our method treats team reward as the causal effect of all the individual rewards and provides a way to recover the underlying parametric model, supported by the theoretical evidence of identifiability. In this way, MACCA offers the ability to distinguish the credit of each agent and gain insights into how their states and actions contribute to the individual rewards and further to the team reward. This is achieved through a learned parameterized generative model that decomposes the team reward into individual rewards. The causal structure within the generative process further enhances our understanding by providing in-

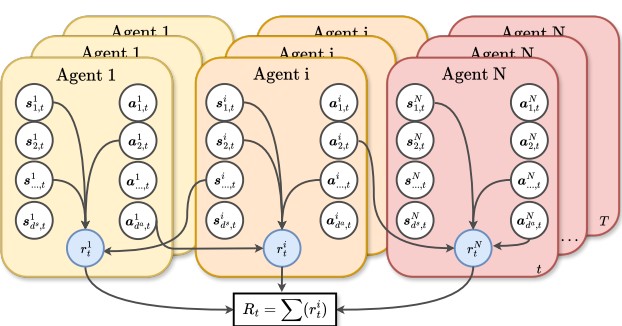

Figure 1: The graphic representation of the causal structure within the MACCA framework. The nodes and edges represent the causal relationships among various environmental variables, i.e., different dimensions of these variables for each agent within the team reward Multi-agent MDP context. These dimensions include the different dimensions of the state $s^i_{\cdots,t}$, action $a^i_{\cdots,t}$, individual reward $r^i_t$ for agent $i$, and the team reward $R_t$. The individual reward $r^i_t$ (shown with blue filling) is unobservable, and the aggregation of $r^i_t$ equals $R_t$.

sights into the specific contributions of each agent. With the support of theoretical identifiability, we identify the unknown causal structure and individual reward function in such a causal generative process. Additionally, our method offers a clear explanation for actions and states leading to individual rewards, promoting policy optimization and invariance. This clarity enhances agent behavior comprehension and aids in refining policies. The inherent modularity of MACCA ensures its compatibility with a range of policy learning methods, positioning it as a versatile and promising MARL solution for various real-world contexts.

We summarize the main contributions of this paper as follows. First, we reformulate team reward decomposition by introducing a Dynamic Bayesian Network (DBN) to describe the causal relationship among states, actions, individual rewards, and team reward. We provide theoretical evidence of identifiability to learn the causal structure and function within the generation of individual rewards and team rewards. Second, our proposed

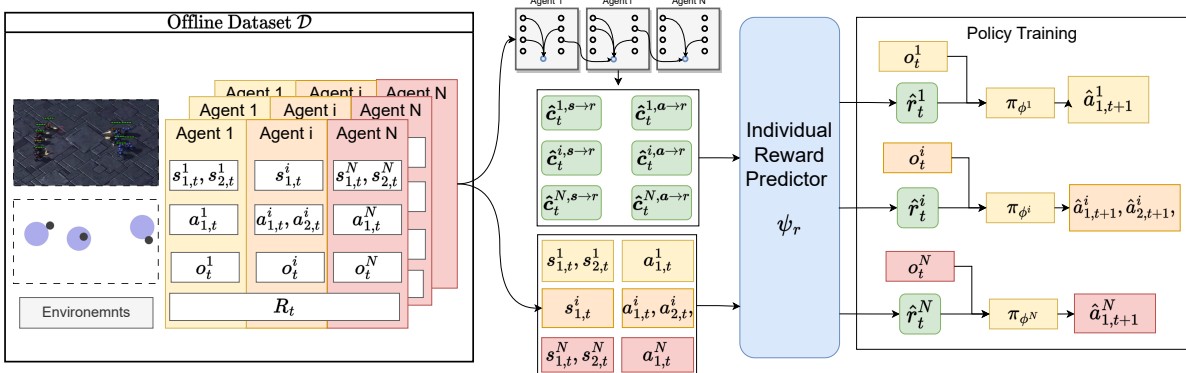

Figure 2: The illustration of the MACCA method. The offline data generation process begins on the left side, where data is recorded from the environment. MACCA then constructs a causal model consisting of a DBN represented in grey and an individual reward predictor depicted in blue. The DBN is used to sample scales from each agent, denoted as $c_t^{i,\cdot\rightarrow\cdot}$ and highlighted in green. Meanwhile, the individual reward predictor takes the joint state, action, and these masks as input to generate the individual reward estimate $\hat{r}_t^i$. During the policy learning phase, each agent utilizes their observation and individual reward estimate as inputs, which are then passed through their respective policy network to generate the next-state actions.

method can recover the parameterized underlying generative process. Lastly, the empirical results on both discrete and continuous action settings, all three environments, demonstrate that MACCA outperforms current state-of-the-art methods in solving the credit assignment problem caused by team rewards.

## 2 Related Work

In this section, we review the close-related topics, *i.e.*, Offline MARL and Multi-agent Credit Assignment and Causal Reinforcement Learning.

**Offline MARL.** Recent research (Pan et al., 2022; Kostrikov et al., 2022; Jiang & Lu, 2021) efforts have delved into offline MARL, identified and addressed some of the issues inherited from offline single-agent RL (Agarwal et al., 2020; Yu et al., 2020; Yang et al., 2022; Wang et al., 2023). For instance, ICQ (Yang et al., 2021) focuses on the vulnerability of multi-agent systems to extrapolation errors, while MABCQ (Jiang & Lu, 2021) examines the problem of mismatched transition distributions in fully decentralized offline MARL. However, these studies all assume using a global state and evaluate the action of the agents relying on the team rewards. Other approaches (Tseng et al., 2022) have a long term progress in online fine-tuning for offline MARL training but have not taken into account the learning slowdown caused by credits of agents to the entire team. For the learning framework, the two most popular recent paradigms are Centralized Training with Decentralized Execution (CTDE) and Independent Learning (IL). Recent research (de Witt et al., 2020; Lyu et al., 2021) shows the benefits of decentralized paradigms, which lead to more robust performance compared to a centralized value function.

**Multi-agent Credit Assignment.** Multi-agent Credit Assignment is the study to decompose the team reward to each individual agent in the cooperative multi-agent environments (Chang et al., 2003; Du et al., 2019; Chen et al., 2023). Recent works (Sunehag et al., 2018; Foerster et al., 2018; Wang et al., 2020; Rashid et al., 2020; Li et al., 2021) focus on value function decompose under online MARL manner. For instance, COMA (Foerster et al., 2018) is a representative method that uses a centralized critic to estimate the counterfactual advantage of an agent action, which is an on-policy algorithm. This means it requires the corresponding data distribution and samples consistent with the current policy for updates. However, in an offline setting, agents are limited to previously collected data and can't interact with the environment. This data, often from varying behavioral policies, might not align with the current policy. Therefore, COMA cannot be directly extended to the offline setting without changing its on-policy features (Levine et al., 2020). In online off-policy settings, state-of-the-art credit assignment algorithms such as SHAQ (Wang et al., 2022a)

and SQDDPG (Wang et al., 2020) utilize an agent's approximate Shapley value for credit assignment. In the experiment section, we conduct a comparative analysis with these methods, and the results for MACCA demonstrate superior performance. Note that we focus on explicitly decomposing the team reward into individual rewards in an offline setting under the casual structure we learned, and these decomposed rewards will be used to reconstruct the offline dataset first and further the policy learning phase. Recent work by Zhang et al. (2024) proposes a causality-inspired spatial-temporal return decomposition approach for multi-agent RL, which further highlights how alternative causal structures can be leveraged to improve credit assignment. However, unlike Zhang et al., their method still assumes access to semi-online or interventional data to disentangle individual contributions, whereas MACCA operates purely on static offline data and comes with a formal identifiability guarantee.

**Causal Reinforcement Learning.** Plenty of work explores solving diverse RL problems with causal structures. Most research emphasizes the transferability of RL agents. For instance, Huang et al. (2021) learn factored representations and domain-specific change factors, while Feng et al. (2022) extend these ideas to handle non-stationary environments. More recently, Wang et al. (2022b) and Pitis et al. (2022) propose removing unnecessary dependencies in causal dynamics models to enhance generalization to unseen states. Hu et al. (2023) exploit causal relationships between actions and reward components to reduce gradient variance during policy learning, and Zhang et al. (2023) leverage causal structures to solve single-agent temporal credit assignment problems. Additionally, causal modeling has been introduced into multi-agent tasks (Grimbly et al., 2021; Jaques et al., 2019), model-based RL (Zhang & Bareinboim, 2016), and imitation learning (Zhang et al., 2020). In the context of games, Hammond et al. (2023) extend Pearl's causal hierarchy (Pearl, 2009b) to game-theoretic scenarios, developing structural causal games to analyze causal effects and counterfactual reasoning among fully observable, rational agents. However, unlike these prior approaches, our method specifically addresses offline settings in partially observable Dec-POMDPs, requiring a causal-identifiability proof to reliably recover individual rewards and causal structures from static datasets. Thus, MACCA complements existing causal RL methods by uniquely targeting causal reward decomposition to facilitate decentralized policy learning under partial observability and offline constraints.

## 3 Preliminaries

In this section, we review the widely-used MARL training framework, the Decentralized Partially Observable Markov Decision Process, and briefly introduce Offline MARL.

**Decentralized Partially Observable Markov Decision Process (Dec-POMDP)** (Oliehoek et al., 2016) is defined by a tuple $\mathcal{M} = \langle N, \mathcal{S}, \mathcal{A}, \mathcal{P}, \mathcal{R}, \mathcal{O}, \gamma \rangle$. In this tuple, $N$ represents the number of agents, $\mathcal{S}$ is the state space, and $\mathcal{A}$ is the shared action spaces and $a^i \in \mathcal{A}$ is the action for agent $i$. The state transition function $\mathcal{P}(s'|s, \boldsymbol{a})$ specifies the probability of transitioning to a new state given the current state $s$ and joint actions $\boldsymbol{a} = (a^1, \ldots, a^N)$. The $R_t = \mathcal{R}(s, \boldsymbol{a})$ is the team reward given by the team reward function and $o^i = \mathcal{O}(s, i)$ is the local observation for agent $i$ at global state $s$. Each agent use a policy $\pi_\theta(a^i|o^i)$ parameterized by $\theta$ to produce an action $a^i$ from the local observation $o^i$, and optimize the discounted accumulated team reward $J_\pi = \mathbb{E}[\sum_{t=0}^{\infty} \gamma^t \mathcal{R}(s_t, \boldsymbol{a}_t)]$, where $\boldsymbol{a}_t = (a_t^1, \ldots, a_t^N)$ is the joint action at time step $t$, and $\gamma$ represents the discount factor.

We assume that the observed team reward $R_t$ can be expressed as the sum of individual rewards, i.e. $R_t = \sum_{i=1}^{N} r_t^i$. Although this does not hold for every cooperative task, it is exactly true (or nearly so) in many common benchmarks (e.g. MPE, SMAC). Importantly, as we will demonstrate later via ablation studies, MACCA remains effective even when the true reward is not strictly additive.

**Offline MARL.** Under offline setting, we consider a MARL scenario where agents sample from a fixed dataset $\mathcal{D} = \{s_t^i, o_t^i, a_t^i, R_t, s_t^{i'}, o_t^{i'}\}$. This dataset is generated from the behavior policy $\pi_b$ without any interaction with the environments, meaning that the dataset is pre-collected offline. Here, $s_t^i$, $o_t^i$ and $a_t^i$ represent the state, observation and action of agent $i$ at time $t$, while $R_t$ is the team reward received at time $t$, and $s_t^{i'}$, $o_t^{i'}$ represents the next state and observation of agent $i$.

**Dynamic Bayesian Networks (DBN) (Murphy, 2002)** is a graphical model for representing the joint distribution over a sequence of random variables across time. A DBN consists of repeated "time slices," where each slice $t$ contains a set of variables (e.g. $X_t$), and edges within the slice encode contemporaneous

dependencies. Between slices, directed edges specify how variables at time $t$ influence variables at time $t+1$, typically under a first-order Markov assumption:

$$P(X_{1:T}) \;=\; P(X_1) \prod_{t=2}^{T} P\big(X_t \mid X_{t-1}\big).$$

Each conditional distribution $P(X_t \mid X_{t-1})$ further factorizes according to the intra-slice structure at time $t$. In this way, a DBN compactly encodes both temporal transitions and within-slice conditional independencies.

## 4 Offline MARL with Causal Credit Assignment

Credit assignment plays a crucial role in facilitating the effective learning of policies in offline cooperative scenarios. In offline MARL, effectively assigning credit among agents is inherently challenging due to partial observability and the static nature of the data. A traditional Decentralized Partially Observable Markov Decision Process (Dec-POMDP) implicitly captures temporal causality; however, it does not explicitly specify which exact dimensions of an agent's state or action affect its individual reward. A generic reward-prediction network—such as a standard regression from joint states and actions directly to a single scalar reward—cannot reveal which agent-specific state-action dimensions drive each agent's contributions, leading to limited interpretability and an absence of sparsity.

To overcome these limitations, we propose MACCA, a method leveraging a causal modeling approach. Specifically, MACCA introduces a Dynamic Bayesian Network (DBN)(Murphy, 2002) to explicitly characterize causal relationships among variables. By learning binary masks, MACCA clearly identifies which subsets of the joint state-action variables causally influence each agent's individual reward, ensuring interpretability and sparsity. Moreover, our DBN formulation guarantees the identifiability of the true causal structure and individual reward functions under standard assumptions—namely the faithfulness and Markov conditions (see Proposition4.1). These critical properties significantly enhance the effectiveness of spatial credit assignment, particularly in scenarios where individual agent rewards are unobservable and further interaction with the environment is impossible.

In the following sections, we first present the underlying generative process within the offline MARL scenario, which serves as the foundation of our method. Then, we describe how to recover the underlying generative process and perform policy learning with the assigned individual rewards.

As shown in Figure 2, our method includes two main components: a causal model $\psi_{\mathrm{m}}$ and a policy model $\psi_{\pi}$. The overall objective, denoted as $L_{\mathrm{MACCA}}$, consists of two key terms: (1) a model estimation loss $L_{\mathrm{m}}$, which measures how accurately the individual reward predictor reconstructs the observed team reward from predicted individual reward components, along with an $\ell_1$ penalty to enforce sparsity in causal masks, and (2) an offline policy learning loss $J_{\pi}$, computed based on standard offline RL algorithms (such as CQL, OMAR, or ICQ), using the assigned individual rewards instead of the team reward. Thus, our combined loss function can be expressed as:

$$L_{\mathrm{MACCA}} = L_{\mathrm{m}} + J_{\pi}, \tag{1}$$

where the specific form of $J_{\pi}$ depends on the offline RL algorithm used (e.g., $J_{\pi}^{\mathrm{CQL}}$, $J_{\pi}^{\mathrm{OMAR}}$, or $J_{\pi}^{\mathrm{ICQ}}$ in this paper).

### 4.1 Underlying Generative Process in MARL

As a foundation of our method, we introduce a Dynamic Bayesian Network (DBN) (Murphy, 2002) to characterize the underlying generative process. DBN is a special type of graphical model that captures the temporal dependencies between variables, corresponding to state transitions across time steps in sequential decision making. By leveraging the DBN structure, we can naturally account for the graph structure over state, action, and reward variables, as well as their temporal dependencies, leading to a natural interpretation of the explicit contribution of each dimension of state and action towards the individual rewards.

We denote the $\mathcal{G}$ as the DBN to represent the causal structure between the states, actions, individual rewards, and team reward as shown in Figure 1, which is constructed over a finite number of random variables as $(s_{1,t}^i, \cdots, s_{d_s^i,t}^i, a_{1,t}^i, \cdots, a_{d_a^i,t}^i, r_t^i, R_t)_{i,t=1}^{N,T}$, where the $d_s^i$ and $d_a^i$ correspond to the dimensions of the state and

action of agent $i$ respectively. $R_t$ is the observed team reward at time step $t$. $r_t^i$ is the unobserved individual reward at time step $t$. $T$ is the maximum episode length of the environment. Then, the underlying generative process is denoted as:

$$\begin{cases} r_t^i = f\big(\boldsymbol{c}^{i,\boldsymbol{s}\to r}\odot\boldsymbol{s}_t,\ \boldsymbol{c}^{i,\boldsymbol{a}\to r}\odot\boldsymbol{a}_t,\ i,\ \varepsilon_{i,t}\big), \\ R_t = \sum_{i=1}^N r_t^i, \end{cases} \tag{2}$$

where $\varepsilon_{i,t}$ denotes an exogenous noise term that is independent and identically distributed across agents and time—i.e. standard structural-causal-model noise—to account for variability beyond the deterministic mapping $f$. Here, $\boldsymbol{s}_t = \{s_{1,t}^1,\dots,s_{d_s^1,t}^1,\dots,s_{1,t}^N,\dots,s_{d_s^N,t}^N\}$ and $\boldsymbol{a}_t = \{a_{1,t}^1,\dots,a_{d_a^1,t}^1,\dots,a_{1,t}^N,\dots,a_{d_a^N,t}^N\}$ are the joint state and joint action of all $N$ agents at time $t$, with $D_{\boldsymbol{s}} = \sum_{i=1}^N d_{\boldsymbol{s}}^i$ and $D_{\boldsymbol{a}} = \sum_{i=1}^N d_{\boldsymbol{a}}^i$ denoting their respective total dimensions. The operator $\odot$ is element-wise multiplication, $f$ is the unknown non-linear individual reward function, and $\boldsymbol{c}^{i,\boldsymbol{s}\to r} \in \{0,1\}^{D_{\boldsymbol{s}}}$, $\boldsymbol{c}^{i,\boldsymbol{a}\to r} \in \{0,1\}^{D_{\boldsymbol{a}}}$ are binary masks (which may be dynamic or static) that indicate which dimensions of $\boldsymbol{s}_t$ and $\boldsymbol{a}_t$ influence $r_t^i$. In particular, if there is a causal edge from the $k$-th dimension of $\boldsymbol{s}_t$ to agent $j$'s individual reward $r_t^j$ in $\mathcal{G}$, then $c^{j,\boldsymbol{s}\to r}(k) = 1$. We emphasize again that we assume an additive structure for the team reward $(R_t = \sum_{i=1}^N r_t^i)$ to facilitate interpretable and computationally efficient credit assignment. As discussed in Section 3, this assumption typically aligns closely with standard cooperative MARL benchmarks, and importantly, MACCA remains effective even when rewards are inherently non-additive.

Each binary mask $c_t^{i,\boldsymbol{s}\to r} \in \{0,1\}^{D_{\boldsymbol{s}}}$ and $c_t^{i,\boldsymbol{a}\to r} \in \{0,1\}^{D_{\boldsymbol{a}}}$ serves two purposes: (1) it reveals the *causal parents* of agent $i$'s individual reward, granting interpretability; and (2) it regularizes $\psi_r$ so that, in an offline dataset with mixed behavior policies, $\psi_r$ does not latch onto spurious correlations among irrelevant state–action dimensions. Without these masks, one would need to predict $\hat{r}_t^i = \psi_r(s_{1:N,t}, a_{1:N,t}, i)$ over all $D_{\boldsymbol{s}} + D_{\boldsymbol{a}}$ dimensions—making it impossible to recover "which features matter" or to prove identifiability. The masks enable both sparsity and a clear DAG structure.

**Proposition 4.1** (Identifiability of Causal Structure and Individual Reward Function). *Suppose the joint state $\boldsymbol{s}_t$, joint action $\boldsymbol{a}_t$, team reward $R_t$ are observable while the individual $r_t^i$ for each agent are unobserved, and they are from the Dec-POMDP, as described in Eq 2. Then under the Markov condition and faithfulness assumption (refer to Appendix C), given the current time step's team reward $R_t$, all the masks $\boldsymbol{c}^{i,\boldsymbol{s}\to r}$, $\boldsymbol{c}^{i,\boldsymbol{a}\to r}$, as well as the function $f$ are identifiable.*

The proposition 4.1 demonstrates that we can identify causal representations from the joint action and state, which serve as the causal parents of the individual reward function we want to fit. This allows us to determine which agent should be responsible for which dimension and thus generate the corresponding individual reward function for each agent. The objective for each agent changes to maximize the sum of individual rewards over an infinite horizon. The proof is in Appendix D.

## 4.2 Causal Model Learning

In this section, we delve into identifying the unknown causal structure and reward function within the graph $\mathcal{G}$. This is achieved using the causal structure predictor $\psi_g$, and the individual reward predictor $\psi_r$. The set $\psi_g = \{\psi_g^{\boldsymbol{s}\to r}, \psi_g^{\boldsymbol{a}\to r}\}$ is to learn the causal structure. Specifically, $\psi_g^{\boldsymbol{s}\to r}$ and $\psi_g^{\boldsymbol{a}\to r}$ are employed to predict the presence of edges in the masks described by Eq 2. We have

$$\hat{\boldsymbol{c}}_t^{i,\boldsymbol{s}\to r} = \psi_g^{\boldsymbol{s}\to r}(\boldsymbol{s}_t,\boldsymbol{a}_t,i), \hat{\boldsymbol{c}}_t^{i,\boldsymbol{a}\to r} = \psi_g^{\boldsymbol{a}\to r}(\boldsymbol{s}_t,\boldsymbol{a}_t,i), \tag{3}$$

where, $\hat{\boldsymbol{c}}_t^{i,\boldsymbol{s}\to r}$ and $\hat{\boldsymbol{c}}_t^{i,\boldsymbol{a}\to r}$ are the predicted masks for agent $i$ at timestep $t$. Note that these causal masks are time-invariant and can change with state and action. We generate masks at each time step since we consider the inherent complexity of the multi-agent scenario, which has high dimensionality and the dynamic nature of the causal relationships that can evolve over time. Thus, we adopt $\psi_g^{\boldsymbol{s}\to r}$ and $\psi_g^{\boldsymbol{a}\to r}$ to generate mask estimation at each time step $t$, within the joint state and joint action and agent id as the input. This dynamic mask adaptation facilitates more accurate causal modelling. To further validate this estimation, we have conducted ablation experiments at Section 5.3.

The $\psi_r$ is used for approximating the function $f$, and is constructed by stacked fully-connection layers. To recover the underlying generative process, i.e., to optimize $\psi_r$, we minimize the following objective:

$$L_{\mathrm{m}} = \mathbb{E}_{\mathcal{D}}[R_t - \sum_{i=1}^{N} \psi_r(\hat{\boldsymbol{c}}_t^{i,\boldsymbol{s}\rightarrow r}, \hat{\boldsymbol{c}}_t^{i,\boldsymbol{a}\rightarrow r}, \boldsymbol{s}_t, \boldsymbol{a}_t, i)]^2 + L_{\mathrm{reg}}. \tag{4}$$

The $L_{\mathrm{reg}}$ serves as an L1 regularization, akin to the purpose delineated in (Zhang & Spirtes, 2011). Its primary objective is to clear redundant features during training, reduce the number of features that a given depends on, and use the coefficients of other features completely set to zero, which fosters model interpretability and mitigates the risk of overfitting. And it defines as:

$$L_{\mathrm{reg}} = \lambda_1 \sum_{i=1}^{N} \|\hat{\boldsymbol{c}}_t^{i,\boldsymbol{s}\rightarrow r}\|_1 + \lambda_2 \sum_{i=1}^{N} \|\hat{\boldsymbol{c}}_t^{i,\boldsymbol{a}\rightarrow r}\|_1, \tag{5}$$

where $\lambda_{(\cdot)}$ are hyper-parameters. For more details, please refer to Appendix F.

### 4.3 Policy Learning with Assigned Individual Rewards.

For policy learning, we use the redistributed individual rewards $\tilde{r}_t^i$ to replace the observed team reward $R_t$. Then, we carry out the policy optimizing over the offline dataset $\mathcal{D}$.

**Individual Rewards Assignment.** We first assign individual rewards for each agent's state-action-id tuple $\langle \boldsymbol{s}_t, \boldsymbol{a}_t, i \rangle$ in the samples used for policy learning. During such an inference phase of individual rewards predictor, we first utilize a hyperparameter, $h$, as an element-wise threshold to determine the existence of the inference phase. Elements within the mask $\hat{\boldsymbol{c}}_t^{i,\boldsymbol{s}\rightarrow r}$ and $\hat{\boldsymbol{c}}_t^{i,\boldsymbol{a}\rightarrow r}$ will be set to 0 if their absolute value is less than h, and 1 otherwise. Then, we assign an individual reward for each agent as,

$$\hat{r}_t^i = \psi_r(\boldsymbol{s}_t, \boldsymbol{a}_t, \hat{\boldsymbol{c}}_t^{i,\boldsymbol{s}\rightarrow r}, \hat{\boldsymbol{c}}_t^{i,\boldsymbol{a}\rightarrow r}, i). \tag{6}$$

**Offline Policy Learning.** The process of individual reward assignment is flexible and is able to be inserted into any policy training algorithm. We now describe three practical offline MARL methods, MACCA-CQL, MACCA-OMAR and MACCA-ICQ. In all those methods, they use Q-Value to guide policy learning, for each agent who estimates the $Q^i(o^i, a^i) = E_\pi[\sum_{t=0}^{\infty} \gamma^t R_t]$ with the Bellman backup operator, we then replace the team reward by learned individual reward $\hat{r}_t^i$ as $\hat{Q}^i(o^i, a^i) = E_\pi[\sum_{t=0}^{\infty} \gamma^t \hat{r}_t^i]$, then in the policy improvement step, MACCA-CQL trains actors by minimizing:

$$J_\pi^{\mathrm{CQL}} = \mathbb{E}_{\mathcal{D}}[(\hat{Q}^i(o^i, a^i) - y^i)^2] + \alpha \mathbb{E}_{\mathcal{D}}[\log \sum_{a^i} \exp(\hat{Q}^i(o^i, a^i)) - \mathbb{E}_{a^i \sim \hat{\pi}_\beta^i}[\hat{Q}^i(o^i, a^i)]], \tag{7}$$

where, $y^i = \hat{r}_t^i + \gamma \min_{k=1,2} \bar{Q}^{i,k}(o^{i'}, \bar{\pi}^i(o^{i'}))$ from Fujimoto et al. (2018) to minimize the temporal difference error, $\bar{Q}^i$ represents the target $\hat{Q}$ for the agent $i$, $\alpha$ is the regularization coefficient, $\hat{\pi}_{\beta^i}$ is the empirical behavior policy of agent $i$ in the dataset. Similarly, MACCA-OMAR updates actors by minimizing:

$$J_\pi^{\mathrm{OMAR}} = -\mathbb{E}_{\mathcal{D}}[(1-\tau)\hat{Q}^i(o^i, \pi^i(o^i)) - \tau(\pi^i(o^i) - \hat{a}_i)^2], \tag{8}$$

where $\hat{a}_i$ is the action provided by the zeroth-order optimizer and $\tau \in [0,1]$ denotes the coefficient. For the MACCA-ICQ, it updates actors by minimizing:

$$J_\pi^{\mathrm{ICQ}} = \mathbb{E}_{\mathcal{D}}[L_2^\tau(\hat{r}(s,a) + \gamma \bar{Q}^i(o^{i'}, a^{i'}) - \hat{Q}^i(o^i, a^i))], \tag{9}$$

where $L_2^\tau$ is the squared loss based on expectile regression and the $\gamma$ is the discount factor, which determines the present value of future rewards. As MACCA uses individual reward to replace the team reward, we do not directly decompose value function, unlike the prior offline MARL methods (Foerster et al., 2018; Wang et al., 2020; 2022a), thus we do not require fitting an additional advantage value or Q-value estimator, simplifying our method.

Table 1: Average Normalized Score of MPE task with Team Reward

| | I-CQL | OMAR | MA-ICQ | MACCA-CQL | MACCA-OMAR | MACCA-ICQ |
|---|---|---|---|---|---|---|
| **Exp(CN)** | 33.6 ± 22.9 | 44.7 ± 46.6 | 45.0 ± 23.1 | 85.4 ± 8.1 | **111.7 ± 4.3** | 90.4 ± 5.1 |
| **Exp(PP)** | 63.4 ± 38.6 | 99.9 ± 14.2 | 87.0 ± 12.3 | 94.9 ± 27.9 | 111.0 ± 21.5 | **114.4 ± 25.1** |
| **Exp(WORLD)** | 54.4 ± 17.3 | 98.7 ± 18.7 | 43.2 ± 15.7 | 89.3 ± 14.8 | **107.4 ± 11.0** | 93.2 ± 12.0 |
| **Med(CN)** | 19.7 ± 8.7 | 49.6 ± 14.9 | 30.8 ± 7.3 | 45.0 ± 8.8 | 67.9 ± 16.9 | **70.3± 10.4** |
| **Med(PP)** | 50.0 ± 15.6 | 57.4 ± 13.9 | 59.4 ± 11.1 | 61.1 ± 27.1 | **87.1 ± 12.2** | 77.4 ± 10.5 |
| **Med(WORLD)** | 25.7 ± 21.3 | 33.4 ± 12.8 | 35.6 ± 6.0 | 54.7 ± 11.0 | **63.6 ± 8.7** | 55.1 ± 3.5 |
| **Med-R(CN)** | 10.8 ± 7.7 | 26.8 ± 15.2 | 22.4 ± 9.3 | 15.9 ± 11.2 | **33.2 ± 12.6** | 28.6 ± 5.6 |
| **Med-R(PP)** | 18.3 ± 9.5 | 56.3 ± 16.6 | 44.2 ± 4.5 | 32.5 ± 15.1 | **69.0 ± 19.3** | 64.3 ± 7.8 |
| **Med-R(WORLD)** | 4.5 ± 10.1 | 28.9 ± 17.2 | 10.7 ± 2.8 | 34.8 ± 16.7 | **50.9 ± 14.2** | 39.9 ± 13.4 |
| **Rand(CN)** | 12.4 ± 9.1 | 22.9 ± 10.4 | 6.0 ± 3.1 | 22.2 ± 4.6 | **32.8 ± 9.5** | 28.13 ± 4.6 |
| **Rand(PP)** | 5.5 ± 2.8 | 12.0 ± 5.2 | 15.6 ± 3.4 | 14.7 ± 6.7 | 20.9 ± 8.3 | **30.3 ± 5.4** |
| **Rand(WORLD)** | 0.1 ± 4.5 | 6.2 ± 6.7 | 0.6 ± 2.4 | 8.7 ± 3.3 | **15.8 ± 6.1** | 10.1 ± 6.6 |

## 5 Experiments

Based on the above, our methods include **MACCA-OMAR**, **MACCA-CQL** and **MACCA-ICQ**. For baselines, we compare with both CTDE and independent learning paradigm methods, including **I-CQL** (Kumar et al., 2020): conservative Q-learning in independent paradigm, **OMAR** (Pan et al., 2022): based on I-CQL, but learning better coordination actions among agents using zeroth-order optimization, **MA-ICQ** (Yang et al., 2021): Implicit constraint Q-learning within CTDE paradigm, **SHAQ** (Wang et al., 2022a) and **SQDDPG** (Wang et al., 2020): variants of credit assignment method using Shapley value, which are the SOTA on the online multi-agent RL, **SHAQ-CQL**: In pursuit of a more fair comparison, we integrated CQL with SHAQ, which adopts the architectural framework of SHAQ while using CQL in the estimations of agents' Q-values and the target Q-values, **QMIX-CQL**: conservative Q-learning within CTDE paradigm, following QMIX structure to calculate the $Q^{tot}$ using a mixing layer, which is similar to the MA-ICQ framework. We evaluate those performance in two environments: Multi-agent Particle Environments (MPE) (Lowe et al., 2017) and StarCraft Micromanagement Challenges (SMAC) (Samvelyan et al., 2019). Through these comparative evaluations, we want to highlight the relative effectiveness and superiority of the MACCA approach. Furthermore, we conduct three ablations to investigate the interpretability and efficiency of our method. For detailed information about the environments, please refer to Appendix E.

### 5.1 General Implementation

**Offline Dataset.** Following the approach outlined in Justin et al. (2020) and Pan et al. (2022), we classify the offline datasets in all environments into four types: Random, generated by random initialization. Medium-Reply, collected from the replay buffer until the policy reaches medium performance. Medium and Expert, collected from partially trained to moderately performing policies and fully trained policies, respectively. The difference between our setup and Pan et al. (2022) is that we hide individual rewards during training and store the sum of these individual rewards in the dataset as the team reward. By creating these different datasets, we aim to explore how different data qualities affect algorithms. For MPE, we adopt the normalized score as a metric to assess performance. The normalized score is calculated by $100 \times (S - S_{\text{random}})/(S_{\text{expert}} - S_{\text{random}})$ following by Justin et al. (2020), where the $S, S_{\text{random}}, S_{\text{expert}}$ are the evaluation return from the current policy, random set behaviour policy, expert set behaviour policy respectively.

### 5.2 Main Results

**Multi-agent Particle Environment (MPE).** We evaluate our method in three distinct environments: Cooperative Navigation (**CN**), Prey-and-Predator (**PP**), and Simple-World (**WORLD**). In the CN environment, three agents aim to reach targets. Observations include position, velocity, and displacements to targets and other agents. Actions are continuous in x and y. Rewards are based on distance to targets, with collision penalties. In the PP environment, three predators chase a random prey. Their state includes position, velocity, and relative displacements. Rewards are based on distance to the prey, with bonuses for captures. The WORLD environment has four allies chasing two faster adversaries. As depicted in Table 1, It can be seen that the algorithms based on MACCA perform better than their respective backbones.

Table 2: Averaged win rate of MACCA-based algorithms and baselines in StarCraft II tasks

| Map | Dataset | I-CQL | OMAR | MA-ICQ | MACCA-CQL | MACCA-OMAR | MACCA-ICQ |
|---|---|---|---|---|---|---|---|
| **2s3z** (Easy) | Expert | 0.70±0.09 | 0.86±0.08 | 0.80±0.01 | 0.88±0.07 | **0.99±0.05** | 0.95±0.01 |
| | Medium | 0.20±0.03 | 0.17±0.01 | 0.16±0.07 | 0.27±0.02 | **0.55±0.03** | 0.51±0.03 |
| | Medium-Replay | 0.11±0.07 | 0.35±0.08 | 0.31±0.04 | 0.25±0.03 | 0.53±0.01 | **0.59±0.04** |
| **5m_vs_6m** (Hard) | Expert | 0.02±0.02 | 0.44±0.04 | 0.38±0.05 | 0.63±0.02 | 0.73±0.04 | **0.88±0.01** |
| | Medium | 0.01±0.00 | 0.14±0.02 | 0.11±0.04 | 0.19±0.01 | **0.20±0.04** | 0.15±0.02 |
| | Medium-Replay | 0.12±0.01 | 0.09±0.04 | 0.18±0.04 | 0.15±0.02 | 0.14±0.01 | **0.28±0.01** |
| **6h_vs_8z** (Super Hard) | Expert | 0.00±0.00 | 0.18±0.08 | 0.04±0.01 | 0.59±0.01 | **0.75±0.07** | 0.60±0.03 |
| | Medium | 0.01±0.01 | 0.12±0.06 | 0.01±0.01 | 0.17±0.00 | 0.20±0.02 | **0.22±0.04** |
| | Medium-Replay | 0.03±0.02 | 0.01±0.01 | 0.07±0.04 | 0.14±0.02 | 0.22±0.01 | **0.25±0.05** |
| **MMM2** (Super Hard) | Expert | 0.08±0.03 | 0.10±0.01 | 0.11±0.01 | 0.60±0.01 | 0.69±0.01 | **0.71±0.03** |
| | Medium | 0.02±0.01 | 0.12±0.02 | 0.08±0.04 | 0.25±0.07 | 0.50±0.06 | **0.59±0.04** |

Table 3: Compare with online off-policy credit assignment baselines in SMAC

| Map | Dataset | SHAQ | SQDDPG | SHAQ-CQL | QMIX-CQL | I-CQL | MACCA-CQL |
|---|---|---|---|---|---|---|---|
| **2s3z** | Expert | 0.10±0.03 | 0.05±0.01 | 0.79±0.03 | 0.73±0.02 | 0.70±0.09 | **0.88±0.07** |
| | Medium | 0.05±0.03 | 0.07±0.01 | 0.24±0.01 | 0.22±0.03 | 0.20±0.03 | **0.27±0.02** |
| **5m_vs_6m** | Expert | 0.02±0.01 | 0.00±0.00 | 0.10±0.03 | 0.03±0.01 | 0.02±0.02 | **0.63±0.02** |
| | Medium | 0.00±0.00 | 0.00±0.00 | 0.06±0.01 | 0.01±0.01 | 0.01±0.00 | **0.19±0.01** |
| **6h_vs_8z** | Expert | 0.00±0.00 | 0.00±0.00 | 0.02±0.01 | 0.00±0.00 | 0.00±0.00 | **0.59±0.01** |
| | Medium | 0.00±0.00 | 0.00±0.00 | 0.04±0.02 | 0.00±0.00 | 0.01±0.01 | **0.17±0.00** |

**StarCraft Micromanagement Challenges (SMAC).** In order to show the performance in the scale scene, we specially selected maps with a large number of agents. To illustrate, the map $2s3z$ needs to control 5 agents, including 2 Stalkers and 3 Zealots, the map 6h_vs_8z needs to control 6 Hydralisks against 8 Zealots, and map MMM2 have 1 Medivac, 2 Marauders and 7 Marines. All experiments will run 3 random seeds and the win rate was recorded, and the corresponding standard was calculated. Table 2 shows the result. For most of the tasks, the MACCA-based method shows state-of-the-art performance compared to their baseline algorithms.

Also, we considered testing online off-policy algorithms in the offline setting. To this end, we introduced several baselines in SMAC for comparison with MACCA, as shown in Table 3. The table below shows the results of the added baselines compared to SMAC tasks. It becomes apparent that when directly applied to the offline setting, online off-policy credit assignment algorithms consistently yield suboptimal performance. Our empirical findings underscore that while SHAQ-CQL indeed exhibits advancements QMIX-CQL, our MACCA-CQL clinches the SOTA performance across all tasks.

### 5.3 Ablation Studies

**The Impact of Learned Causal Structure.** We varied the value of $\lambda_1$ in Eq 5 to control the density of the learned causal structure. Table 4 presents the average cumulative reward and the density of the causal structure during the training process in the MPE-CN environment. The density of the causal structure $\hat{c}_t^{i,s\rightarrow r}$, is calculated as $\rho_{sr} = \sum_{i=1}^{N} \frac{1}{d_s^i} \sum_{k=1}^{d_s^i} s_k^{i,s\rightarrow r}$, where $s_k^{i,s\rightarrow r}$ represent is the value bigger than the threshold $h$. The results indicate that as $\lambda_1$ increases from 0 to 0.5, the causal structure becomes more sparse ($\rho_{sr}$ decreases), resulting in less policy improvement. This can be attributed to the fact that MACCA may not have enough states to predict individual rewards, leading to misguided policy learning accurately. Conversely, setting a relatively low $\lambda_1$ may result in a denser structure that incorporates redundant dimensions, hindering policy learning. Therefore, achieving a reasonable causal structure for the reward function can improve both the convergence speed and the performance of policy training. We also provide the ablation for $\lambda_2$, please refer to Appendix.F.4.

**Ground Truth Individual Reward.** In the MPE-CN expert dataset, we investigate the influence of ground truth individual rewards on agent policy updates. Two scenarios are compared: agents update policies using ground truth individual rewards (GT), and agents primarily rely on team rewards (without GT).

Table 5: Average normalized scores for ground truth individual reward comparison in MPE-CN

| | OMAR | MACCA-OMAR |
|---|---|---|
| With GT | 114.9 ± 2.4 | 113.7 ± 2.3 |
| Without GT | 43.7 ± 46.6 | 111.7 ± 4.3 |

Notably, OMAR with GT directly employs individual rewards for policy updates, while MACCA-OMAR with

Table 4: The mean and the standard variance of average normalized score, density rate $\rho_{sr}$ of $\hat{c}_t^{i,s\to r}$ with diverse $\lambda_1$ at different time step $t$ in MPE-CN.

| $\lambda_1$ / $t$ | 1e4 | 3e4 | 5e4 | 1e5 | 2e5 |
|---|---|---|---|---|---|
| 0 | -2.43 ± 8.01(0.98) | -14.87± 7.71(0.90) | -12.356± 5.83(0.81) | 9.842± 18.89(0.77) | 69.04 ± 19.69(0.72) |
| 0.007 | -7.88±5.36(0.94) | **13.26±27.14(0.47)** | **60.18±26.14(0.28)** | **99.78± 19.50(0.15)** | **111.65± 4.28(0.13)** |
| 0.05 | -3.66±12.14(0.90) | 3.93±42.06(0.34) | 10.04± 45.97(0.17) | 23.61± 44.18(0.11) | 75.81± 34.48(0.10) |
| 0.5 | -12.20±3.87(0.87) | -16.19±5.53(0.24) | -8.84± 7.16(0.11) | 16.40± 21.04(0.07) | 59.23± 35.29(0.01) |

GT utilizes individual rewards as a supervisory signal, replacing team rewards in Eq 4. The results, presented in Table 5, the small gap between MACCA+GT and OMAR+GT arises from the extra regularization introduced by jointly learning $\psi_g$ and $\psi_r$, a design choice that in fact improves generalization when ground truth $r_t^i$ are unavailable. Thus, even though OMAR with GT slightly outperforms MACCA–OMAR when GT is provided, MACCA–OMAR's ability to learn $\hat{r}_t^i$ without supervision is what enables major performance gains over baselines that rely solely on team rewards. For further details on prediction accuracy convergence versus GT, see Appendix F.6

**The Impact of Causal Graph Types.** To investigate the performance under different graph types, we consider three settings. The Fully Connected Graph assumes all variables are causally connected, while The Fixed Graph learns a static graph that is invariant to time by averaging the predicted masks $\hat{c}_t^{i,\cdot\to r}$ overall time steps during training. Our proposed graph setting, as described in Equation 3, learns a graph that depends on the current state $s_t$ and action $a_t$. Table 6 presents the results of MACCA-OMAR under these different graph types. The Fully Connected Graph yields suboptimal performance due to its inability to differentiate individual agent contributions. The Fixed Graph shows marginal improvement over the Fully Connected Graph but remains limited in capturing the complex dynamic multi-agent causal relationships that vary with time. In contrast, our proposed dynamic graph setting achieves the highest performance by incorporating time-varying information. Additionally, we compared the performance of our method with and without $h$ clipping, where the threshold $h$ filters the causal mask. The results demonstrate that our method with $h$ clipping outperforms the variant without it. This aligns with established practices in earlier works on DAG structural learning (Zheng et al., 2018; Ng et al., 2020), which show the importance of clipping to ensure edge weights converge to zero when working with finite datasets. Appendix F.5 provides additional results of MACCA under different levels of $h$.

Table 6: Average win rate in SMAC 5m_vs_6m map, expert dataset.

| | Win Rate |
|---|---|
| MACCA (Fully Connected Graph) | 0.38 ± 0.02 |
| MACCA (Fixed Graph) | 0.50 ± 0.01 |
| MACCA (w.o $h$ clipping) | 0.66 ± 0.01 |
| MACCA (w. $h$ clipping) | **0.73 ± 0.04** |

**Visualization of Causal Structure.** In Figure 3, we provide visualizations of two significant causal structures within the CN environment of MPE. To observe the causal structure learning process more easily, we initialize the $\hat{c}_t^{i,s\to r}$ as a normalized random number close to 1 and the $\hat{c}_t^{i,a\to r}$ close to 0. Over time, we notice that the causal structure $\hat{c}_t^{i,s\to r}$ shifts its focus from considering all dimensions of the agent state to primarily emphasizing the $4^{th}$ to $10^{th}$ dimensions of each agent. In this environment, the agent's state comprises 18 dimensions. Specifically, dimensions $0^{th}$ to $4^{th}$ us agent's velocity and position, $5^{th}$ to $9^{th}$ capture the distance between the agent and three distinct landmarks, $10^{th}$ to $13^{th}$ reflect the distances between the agent and other agents, and dimensions $14^{th}$ to $17^{th}$ are related to communication, although not applicable in this experiment and thus considered irrelevant. In other words, the dimensions $4^{th}$ to $9^{th}$ and $10^{th}$ to $13^{th}$ are intuitively linked to individual rewards, aligning with the convergence direction of MACCA. With regard to the causal structure $\hat{c}_t^{i,a\to r}$, as each agent's actions involve continuous motion without extraneous variables, it converges to relevant states that contribute to individual credits for the team reward. The results support the interpretability of relationships between variables through the causal structure.

**Training paradigms** In MACCA, we train the causal model and policy alternately rather than train the causal model at the beginning. The benefit of alternated training is that the reward model is less accurate at the early stage of training, which encourages agents to extract diverse behaviours that go beyond the dataset. Similar to (Hu et al., 2024), they discuss the usefulness of random rewards prior. We conducted experiments as detailed in Table 7. Here, the **TCB** stands for training the causal model at the

Table 7: The win rate and loss of different training paradigms by using MACCA-OMAR in SMAC 5m_vs_6m, expert dataset

| | Win Rate | Causal Model Loss |
|---|---|---|
| TCB | 0.62 ± 0.08 | 0.80 ± 0.02 |
| TCPA | 0.73 ± 0.04 | 0.81 ± 0.01 |

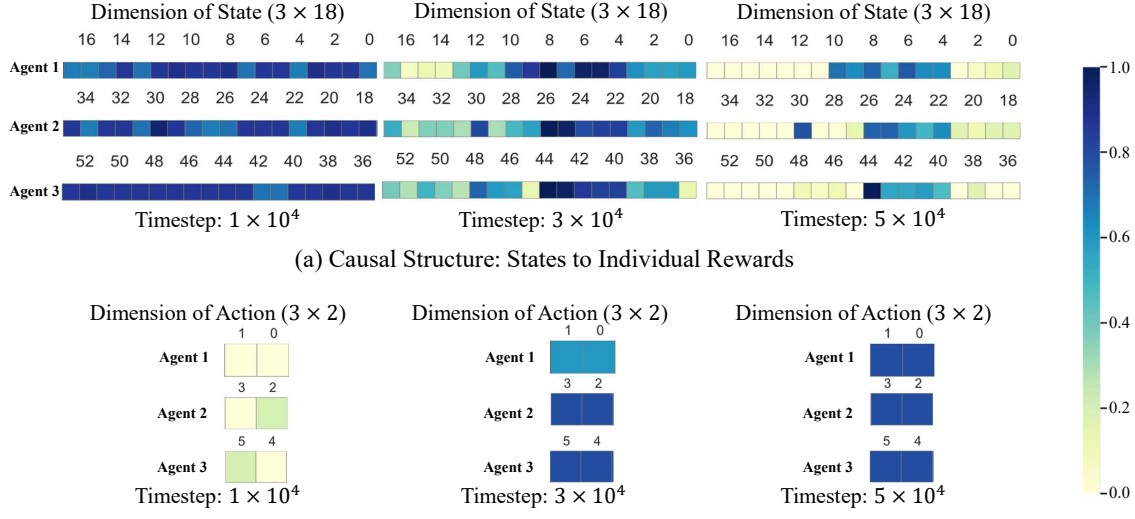

(a) Causal Structure: States to Individual Rewards

(b) Causal Structure: Actions to Individual Rewards

Figure 3: The figure visualizes the causal structure, showing the probability of causal edges from blue (high probability) to yellow (low probability). **(a)** represents the causal structure $\hat{c}_t^{i,s \to r}$ between the state of all agents (18 dimensions for each agent, 54 dimensions for joint state ) and the individual reward (1 dimension for each agent). **(b)** represents the causal structure $\hat{c}_t^{i,a \to r}$ between the action of each agent (2 dimensions for each agent, six dimensions for joint action) and the individual reward (1 dimension for each agent).

beginning, and the **TCPA** is training the causal model and policy alternately. The causal model is initially trained with the same training time steps as the alternating setting, which is 10 million steps. According to the result, for both paradigms, the reward model loss converged to comparable levels, and TCPA showed a clear improvement in the win rate.

**Robustness to Non-Additive Rewards.** On a sparse-reward variant of MPE-CN—where $R_t = 1$ only if all three agents cover distinct landmarks and $R_t = 0$ otherwise (so $\sum_i r_t^i \neq R_t$)—MACCA still outperforms the baseline (Table 8), demonstrating that it can learn meaningful latent individual rewards and effective policies even when the team reward is not decomposable.

Table 8: Performance on the sparse-reward variant of MPE-CN. Despite the highly non-additive team reward, MACCA significantly improves credit assignment and overall policy performance.

| Method | Avg. Episode Reward |
|---|---|
| OMAR | $0.18 \pm 0.07$ |
| MACCA-OMAR | $0.42 \pm 0.13$ |

## 6 Conclusion

In conclusion, MACCA emerges as a valuable solution to the credit assignment problem in offline Multi-agent Reinforcement Learning (MARL), providing an interpretable and modular framework for capturing the intricate interactions within multi-agent systems. By leveraging the inherent causal structure of the system, MACCA allows us to disentangle and identify the specific credits of individual agents to team rewards. This enables us to accurately assign credit and update policies accordingly, leading to enhanced performance compared to different baseline methods. The MACCA framework empowers researchers and practitioners to gain deeper insights into the dynamics of multi-agent systems, facilitating the understanding of the causal factors that drive cooperative behavior and ultimately advancing the capabilities of MARL in a variety of real-world applications.

**Limitation and Future Work.** One limitation of the current work is that the experiments focused on simulated environments rather than real-world scenarios. While the MPE and SMAC environments provide controlled testbeds to evaluate the approach, the performance of MACCA in practical multi-agent applications remains to be investigated. Future work could explore integrating the method with real robot systems or testing it on datasets collected from real-world multi-agent interactions to further validate its practicality and robustness.

## Acknowledge

This work was supported by the Engineering and Physical Sciences Research Council [grant number EP/Y003187/1]

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

# A   Broader Impact Statements

The proposed work advances offline multi-agent reinforcement learning by introducing a general approach to credit assignment that can be integrated with existing algorithms, yielding significant performance gains in static-data settings. Beyond academic benchmarks, MACCA's identifiable causal structures offer critical value in real-world applications: for instance, multi-robot coordination in logistics or disaster-response teams—where understanding each robot's contribution is essential for fault diagnosis and task allocation; autonomous driving fleets—where clear, interpretable attributions of responsibility among vehicles improve accountability, safety, and regulatory compliance; and distributed control systems in domains such as smart grids or algorithmic trading—where transparent, causally grounded decisions enhance reliability and risk management. At the same time, we recognize risks inherent in learning causal models from biased or incomplete offline datasets—misestimated causal relations could lead to unfair credit assignment or unsafe behavior in high-stakes environments. To mitigate these risks, we emphasize rigorous dataset auditing to detect sampling biases, validation of learned causal structures through simulation before deployment, and cautious incremental rollout in mission-critical applications to ensure that decisions remain safe and equitable."'

# B   Reproducibility Statements

To promote transparent and accountable research practices, we have prioritized the reproducibility of our method. All experiments conducted in this study adhere to controlled conditions and well-known environments and datasets, with detailed descriptions of the experimental settings available in Section 5 and Appendix E. The implementation specifics for all the baseline methods and our proposed MACCA are thoroughly outlined in Section 4 and Appendix F.

# C   Markov and Faithfulness Assumptions

A directed acyclic graph (DAG), $\mathcal{G} = (\boldsymbol{V}, \boldsymbol{E})$, can be deployed to represent a graphical criterion carrying out a set of conditions on the paths, where $\boldsymbol{V}$ and $\boldsymbol{E}$ denote the set of nodes and the set of directed edges, separately.

**Definition C.1.** (d-separation (Pearl, 2009a)). A set of nodes $\boldsymbol{Z} \subseteq \boldsymbol{V}$ blocks the path $p$ if and only if (1) $p$ contains a chain $i \to m \to j$ or a fork $i \leftarrow m \to j$ such that the middle node $m$ is in $\boldsymbol{Z}$, or (2) $p$ contains a collider $i \to m \leftarrow j$ such that the middle node $m$ is not in $\boldsymbol{Z}$ and such that no descendant of $m$ is in $\boldsymbol{Z}$. Let $\boldsymbol{X}$, $\boldsymbol{Y}$ and $\boldsymbol{Z}$ be disjoint sets of nodes. If and only if the set $\boldsymbol{Z}$ blocks all paths from one node in $\boldsymbol{X}$ to one node in $\boldsymbol{Y}$, $\boldsymbol{Z}$ is considered to d-separate $\boldsymbol{X}$ from $\boldsymbol{Y}$, denoting as $(\boldsymbol{X} \perp\!\!\!\perp_d \boldsymbol{Y} \mid \boldsymbol{Z})$.

**Definition C.2.** (Global Markov Condition (Spirtes et al., 2000; Pearl, 2009a)). If, for any partition $(\boldsymbol{X}, \boldsymbol{Y}, \boldsymbol{Z})$, $\boldsymbol{X}$ is d-separated from $\boldsymbol{Y}$ given $\boldsymbol{Z}$, i.e. $\boldsymbol{X} \perp\!\!\!\perp_d \boldsymbol{Y} \mid \boldsymbol{Z}$. Then the distribution $P$ over $\boldsymbol{V}$ satisfies the global Markov condition on graph $G$, and can be factorizes as, $P(\boldsymbol{X}, \boldsymbol{Y} \mid \boldsymbol{Z}) = P(\boldsymbol{X} \mid \boldsymbol{Z})P(\boldsymbol{Y} \mid \boldsymbol{Z})$. That is, $\boldsymbol{X}$ is conditionally independent of $\boldsymbol{Y}$ given $\boldsymbol{Z}$, writing as $\boldsymbol{X} \perp\!\!\!\perp \boldsymbol{Y} \mid \boldsymbol{Z}$.

**Definition C.3.** (Faithfulness Assumption (Spirtes et al., 2000; Pearl, 2009a)). The variables, which are not entailed by the Markov Condition, are not independent of each other.

Under the above assumptions, we can apply d-separation as a criterion to understand the conditional independencies from a given DAG $G$. That is, for any disjoint subset of nodes $\boldsymbol{X}, \boldsymbol{Y}, \boldsymbol{Z} \subseteq \boldsymbol{V}$, $(\boldsymbol{X} \perp\!\!\!\perp \boldsymbol{Y} \mid \boldsymbol{Z})$ and $\boldsymbol{X} \perp\!\!\!\perp_d \boldsymbol{Y} \mid \boldsymbol{Z}$ are the necessary and sufficient condition of each other.

# D   Proof of Identifiability

**Proposition D.1** (Individual Reward Function Identifiability)**.** *Suppose the joint state $\boldsymbol{s}_t$, joint action $\boldsymbol{a}_t$, team reward $R_t$ are observable while the individual $r_t^i$ for each agent are unobserved, and they are from the Dec-POMDP, as described in Eq 2. Then, under the Markov condition and faithfulness assumption, given the current time step's team reward $R_t$, all the masks $\boldsymbol{c}^{\boldsymbol{s} \to r,i}$, $\boldsymbol{c}^{\boldsymbol{a} \to r,i}$, as well as the function $f$ are identifiable.*

**Assumption**   We assume that, $\epsilon_{i,t}$ in Eq 2 are i.i.d additive noise. From the weight-space view of Gaussian Process (Williams & Rasmussen, 2006) and equation.6, equivalently, the causal models for $r_t^i$ can be represented as follows,

$$r_t^i = f(\boldsymbol{c}_t^{i,\boldsymbol{s} \to r} \odot \boldsymbol{s}_t, \boldsymbol{c}_t^{i,\boldsymbol{a} \to r} \odot \boldsymbol{a}_t, i) + \epsilon_{r,t} = W_f{}^T \phi_r(\boldsymbol{s}_t, \boldsymbol{a}_t, i) + \epsilon_{i,t} \tag{10}$$

where $\forall i \in [1, N]$, and $\phi_r$ denote basis function sets.

As $\boldsymbol{s}_t = \{s^1_{1,t}, ..., s^1_{d^1_s,t}, ..., s^N_{1,t}, ..., s^N_{d^N_s,t}\}$ and $\boldsymbol{a}_t = \{a^1_{1,t}, ..., a^1_{d^1_a,t}, ..., a^N_{1,t}, ..., a^N_{d^N_a,t}\}$. We denote the variable set in the system by $\boldsymbol{V} = \{\boldsymbol{V}_0, ..., \boldsymbol{V}_T\}$, where $\boldsymbol{V}_t = \boldsymbol{s}_t \cup \boldsymbol{a}_t \cup R_t$, and the variables form a Bayesian network $\mathcal{G}$. Following AdaRL (Huang et al., 2021), there are possible edges only from $s^i_{k,t} \in \boldsymbol{s}_t$ to $r^i_t$, and from $a^i_{j,t} \in \boldsymbol{a}_t$ to $r^i_t$ in $\mathcal{G}$, where $k, j$ are dimension index in $[1, ..., d^N_s]$ and $[1, ..., d^N_a]$ respectively. In particular, the $r^i_t$ are unobserved, while $R_t = \sum_{i=1}^N r^i_t$ is observed. Thus, there are deterministic edges from each $r^i_t$ to $R_t$.

**Proof of the Proposition B.1** We aim to prove that, given the team reward $R_t$, and the $\boldsymbol{c}^{i,\boldsymbol{s}\to r}$, $\boldsymbol{c}^{i,\boldsymbol{a}\to r}$ and $r^i_t$ are identifiable. Following the above assumption, we can rewrite the Eq 2 to the following,

$$
\begin{aligned}
R_t &= \sum_{i=1}^N r^i_t \\
&= \sum_{i=1}^N \left[ W_f{}^T \phi_r(\boldsymbol{s}_t, \boldsymbol{a}_t, i) + \epsilon_{i,t} \right] \\
&= W_f{}^T \sum_{i=1}^N \phi_r(\boldsymbol{s}_t, \boldsymbol{a}_t, i) + \sum_{i=1}^N \epsilon_{i,t}.
\end{aligned}
\tag{11}
$$

For simplicity, we replace the components in Eq 11 by,

$$
\begin{aligned}
\Phi_{r,t} &= \sum_{i=1}^N \phi_r(\boldsymbol{s}_t, \boldsymbol{a}_t, i), \\
\mathcal{E}_{r,t} &= \sum_{i=1}^N \epsilon_{i,t}.
\end{aligned}
\tag{12}
$$

Consequently, we derive the following equation,

$$
R_t = W_f{}^T \Phi_{r,t}(X_t) + \mathcal{E}_{r,t},
\tag{13}
$$

where $X_t := [\boldsymbol{s}_t, \boldsymbol{a}_t, i]_{i=1}^N$ representing the concatenation of the covariates $\boldsymbol{s}_t$, $\boldsymbol{a}_t$ and $i$, from $i = 1$ to $N$.

Then we can obtain a closed-form solution of $W_f{}^T$ in Eq 13 by modelling the dependencies between the covariates $X_t$ and response variables $R_t$. One classical approach to finding such a solution involves minimizing the quadratic cost and incorporating a weight-decay regularizer to prevent overfitting. Specifically, we define the cost function as,

$$
C(W_f) = \frac{1}{2} \sum_{X_t, R_t \sim \mathcal{D}} (R_t - W_f{}^T \Phi_{r,t}(X_t))^2 + \frac{1}{2} \lambda \|W_f\|^2.
\tag{14}
$$

where $X_t$ and long-term returns $R_t$, which are sampled from the offline dataset $\mathcal{D}$. $\lambda$ is the weight-decay regularization parameter. To find the closed-form solution, we differentiate the cost function with respect to $W_f$ and set the derivative to zero:

$$
\frac{\partial C(W_f)}{\partial W_f} \to 0.
\tag{15}
$$

Solving Eq 15 will yield the closed-form solution for $W_f$, as

$$
W_f = (\lambda I_d + \Phi_{r,t} \Phi_{r,t}{}^T)^{-1} \Phi_{r,t} R_t = \Phi_{r,t} (\Phi_{r,t}{}^T \Phi_{r,t} + \lambda I_n)^{-1} R_t.
\tag{16}
$$

Therefore, $W_f$, which indicates the causal structure and strength of the edge, can be identified from the observed data. In summary, given team reward $R_t$, the binary masks, $\boldsymbol{c}^{i,\boldsymbol{s}\to r}$, $\boldsymbol{c}^{i,\boldsymbol{a}\to r}$ and individual $r^i_t$ are identifiable.

Considering the Markov condition and faithfulness assumption, we can conclude that for any pair of variables $V_k, V_j \in \mathbf{V}$, $V_k$ and $V_j$ are not adjacent in the causal graph $\mathcal{G}$ if and only if they are conditionally independent given some subset of $\{V_l \mid l \neq k, l \neq j\}$. Additionally, since there are no instantaneous causal relationships and the direction of causality can be determined if an edge exists, the binary structural masks $\mathbf{c}^{i,\mathbf{s}\rightarrow r}$ and $\mathbf{c}^{i,\mathbf{a}\rightarrow r}$ defined over the set $\mathbf{V}$ are identifiable with conditional independence relationships (Huang et al., 2022). Consequently, the functions $f$ in Equation 2 are also identifiable.

## E   Environments Setting

We adopt the open-source implementations for the multi-agent particle environment (Lowe et al., 2017)[1] and SMAC(Samvelyan et al., 2019)[2]. The tasks in the multi-agent particle environments are illustrated in Figures 4(a)-(c). The Cooperative Navigation (CN) task involves 3 agents and 3 landmarks, requiring agents to cooperate in covering the landmarks without collisions. In the Predator-Prey (PP) task, 3 predators collaborate to capture prey that is faster than them. Finally, the WORLD task features 4 slower cooperating agents attempting to catch 2 faster adversaries, with the adversaries aiming to consume food while avoiding capture.

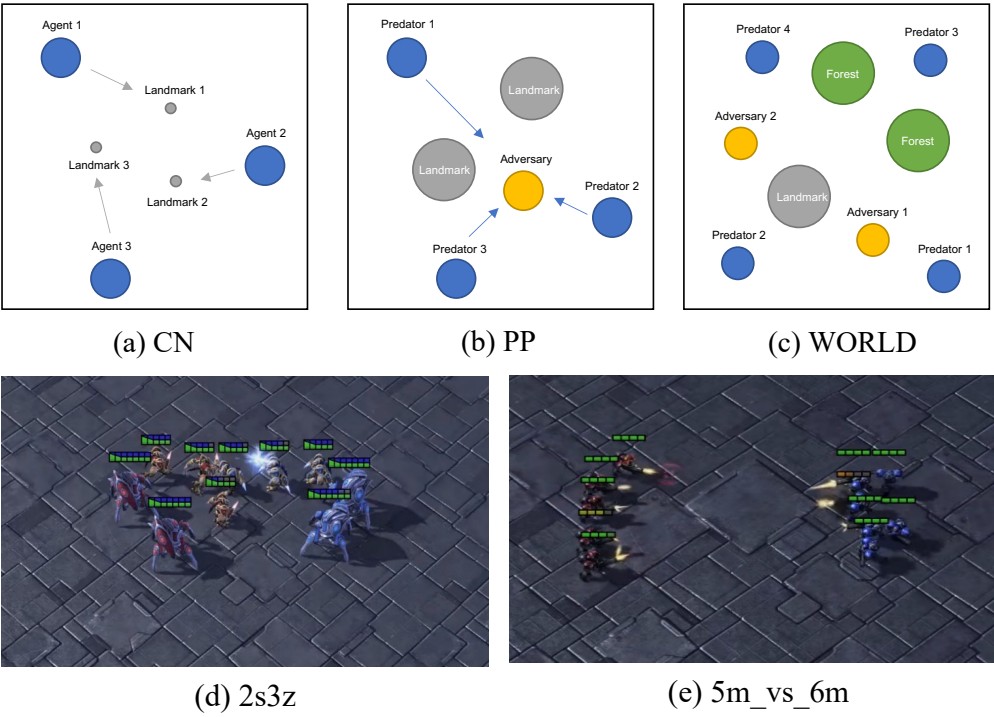

(a) CN          (b) PP          (c) WORLD

(d) 2s3z                    (e) 5m_vs_6m

Figure 4: Visualization of different environment in the experiments, **(a)-(c):** Multi-agent Particle Environments (MPE), **(d)-(e):** StarCraft Micromanagement Challenges (SMAC)

**Datasets.** During training, we utilize the team reward as input, while for evaluation purposes, we compare the performance with the ground truth individual reward. As a result, the expert and random scores for the Cooperative Navigation, Predator-Prey and World tasks are as follows: Cooperative Navigation - expert: 516.526, random: 160.042; Predator-Prey - expert: 90.637, random: -2.569; World - expert: 34.661, random: -8.734;

---

[1]https://github.com/openai/multiagent-particle-envs
[2]https://github.com/oxwhirl/smac

# F   Implementations

## F.1   Algorithm

---
**Algorithm 1** MACCA: **M**ulti-**A**gent **C**ausal **C**redit **A**ssignment
---
1: **for** training step $t = 1$ to $T$ **do**
2:     Sample trajectories from $\mathcal{D}$, save in minibatch $\mathcal{B}$
3:     **for** agent $i = 1$ to $N$ **do**
4:         Update the team reward $R_t$ to $\hat{r}_t^i$ in $\mathcal{B}$ (Eq 6)
5:         Optimize $\psi_m$: $\psi_m \leftarrow \psi_m - \alpha \nabla_{\psi_m} L_m$ (Eq 4)
6:     Update policy $\pi$ with minibatch $\mathcal{B}$ (Eq 7, Eq 8 or Eq 9)
7:     Reset $\mathcal{B} \leftarrow \emptyset$
---

## F.2   Model Structure

The parametric generative model $\psi_m$ used in MACCA consists of two parts: $\psi_g$ and $\psi_r$. The function of $\psi_g$ is to predict the causal structure, which determines the relationships between the environment variables. The role of $\psi_r$ is to generate individual rewards based on the joint state and action information. This prediction is achieved through a network architecture that includes three fully-connected layers with an output size of 256, followed by an output layer with a single output. Each hidden layer is activated using the rectified linear unit (ReLU) activation function.

During the training process, the generative model is optimized to learn the causal structure and generate individual rewards that align with the observed team rewards. The model parameters are updated using Adam, to minimize the discrepancy between the predicted sum of individual rewards and the team rewards. The training process involves iteratively adjusting the parameters to improve the accuracy of the predictions.

For a more detailed overview of the training process, including the specific loss functions and optimization algorithms used, please refer to Figure 2. The Figure provides a step-by-step illustration of the training pipeline, helping to visualize the flow of information and the interactions between different components of the generative model.

Table 9: The Common Hyperparameters.

| hyperparameters | value | hyperparameters | value |
|---|---|---|---|
| steps per update | 100 | optimizer | Adam |
| batch size | 1024 | learning rate | $3 \times 10^{-4}$ |
| hidden layer dim | 64 | $\gamma$ | 0.95 |
| evaluation interval | 1000 | evaluation episodes | 10 |

Table 10: Hyperparameters for OMAR, CQL and MACCA

| | OMAR $\tau$ | CQL $\alpha$ | MACCA $\lambda_1$ | MACCA $\lambda_2$ | MACCA $r_{lr}$ | MACCA $h$ |
|---|---|---|---|---|---|---|
| Expert | 0.9 | 5.0 | 7e-3 | 7e-3 | 5e-2 | 0.1 |
| Medium | 0.7 | 0.5 | 5e-3 | 5e-3 | 5e-2 | 0.1 |
| Medium-Replay | 0.7 | 1.0 | 5e-3 | 7e-3 | 5e-2 | 0.1 |
| Random | 0.99 | 1.0 | 1e-7 | 1e-3 | 5e-2 | 0.1 |

## F.3   Hyper-parameters

The common hyperparameters are shown in Table.9. The neural network used in training is initialized from scratch and optimized using the Adam optimizer with a learning rate of $3 \times 10^{-4}$. The policy learning process involves varying initial learning rates based on the specific algorithm, while the hyperparameters for policy learning, including a discount factor of 0.95, are consistent across all tasks.

The training procedure differs across tasks. For MPE, the training duration ranges from 20,000 to 60,000 iterations, with longer training for behavior policies that perform poorly. The number of steps per update is set to 100.

During each training iteration, trajectories are sampled from the offline data, and the generated individual reward is replaced with the team reward for policy updates. The training of $\psi_{\text{cau}}$ is performed concurrently with $\psi_{\text{rew}}$. Validation is conducted after each epoch, and the average metrics are computed using 5 random seeds for reliable evaluation.

The hyperparameters specific to training MACCA models can be found in Table 10. All experiments were conducted on a high-performance computing (HPC) system featuring 128 Intel Xeon processors running at 2.2 GHz, 5 TB of memory, and an Nvidia A100 PCIE-40G GPU. This computational setup ensures efficient processing and reliable performance throughout the experiments.

### F.4 Ablation for $\lambda_2$

We have conducted ablation experiments on $\lambda_2$ and show the results in the Table 11.

Table 11: The mean and the standard variance of average normalized score, sparsity rate $\rho_{ar}$ of $\hat{c}_t^{i, \boldsymbol{a} \to r}$ with diverse $\lambda_2$ at different time step $t$ in MPE-CN.

| $\lambda_2$ / $t$ | 1e4 | 5e4 | 1e5 | 2e5 |
|---|---|---|---|---|
| 0 | $17.4 \pm 15.2(0.98)$ | $93.1 \pm 6.4\ (1.0)$ | $105 \pm 3.5\ (1.0)$ | $107.7 \pm 10.2\ (1.0)$ |
| 0.007 | $19.9 \pm 12.4\ (0.8$ | $\mathbf{90.2 \pm 7.1\ (1.0)}$ | $\mathbf{108.8 \pm 4.0\ (1.0)}$ | $\mathbf{111.7 \pm 4.3(1.0)}$ |
| 0.5 | $13.3 \pm 11.1\ (0.68)$ | $100.5 \pm 14.0\ (0.84)$ | $102.9 \pm 16.4\ (0.87)$ | $108.4 \pm 6.4\ (0.98)$ |
| 5.0 | $2.3 \pm 9.8\ (0.0)$ | $-1.3 \pm 25.4\ (0.34)$ | $70.4 \pm 18.0\ (0.62)$ | $100.1 \pm 7.4\ (0.75)$ |

### F.5 Ablation for $h$

The selection of $h$ can influence the sparsity of the causal graph. $h$ can be selected by parameter sweeping. For simplicity, we use $h = 0.1$ for all tasks in the experiments, which leads to strong performance. we conduct additional experiments under different $h$ in SMAC 5m_vs_6m Medium Dataset with MACCA-OMAR. The results are as follows,

Table 12: The mean and the standard variance of the average normalized score, sparsity rate $\rho_{ar}$ of $\hat{c}_t^{i, \boldsymbol{a} \to r}$ with diverse $h$ in SMAC 5m_vs_6m.

| $h$ | Win Rate | $\rho_{sr}$ | $\rho_{ar}$ | Causal Model Loss |
|---|---|---|---|---|
| 0 | $0.12 \pm 0.02$ | $1.0 \pm 0.0$ | $1.0 \pm 0.0$ | $0.15 \pm 0.05$ |
| 0.01 | $0.14 \pm 0.03$ | $0.96 \pm 0.12$ | $0.72 \pm 0.12$ | $0.07 \pm 0.01$ |
| 0.05 | $0.16 \pm 0.02$ | $0.81 \pm 0.07$ | $0.66 \pm 0.04$ | $0.09 \pm 0.04$ |
| 0.1 | $0.20 \pm 0.04$ | $0.73 \pm 0.04$ | $0.54 \pm 0.08$ | $0.05 \pm 0.02$ |
| 0.5 | $0.17 \pm 0.01$ | $0.52 \pm 0.10$ | $0.43 \pm 0.07$ | $0.12 \pm 0.06$ |

The causal graph become more sparse (fewer edges between nodes) with the increase of $h$. The performance of win rate goes up with the increase of $h$ but decrease after $h > 0.1$, due to potential inclusion of redudance information.

### F.6 Prediction Accuracy After Convergence (By Dataset Quality)

After convergence, we measure per-agent errors between redistributed rewards $\hat{r}_t^i$ and true $r_t^i$. Table 13 shows MSE and MAE (averaged over time and three random seeds) for Expert, Medium, Medium-Replay, and Random datasets.

Table 13: Per-Agent Prediction Errors After Convergence

| Dataset Quality | MSE | MAE |
|---|---|---|
| Expert | $0.12 \pm 0.02$ | $0.65 \pm 0.10$ |
| Medium | $0.24 \pm 0.05$ | $1.12 \pm 0.15$ |
| Medium-Replay | $0.38 \pm 0.08$ | $1.78 \pm 0.20$ |
| Random | $1.60 \pm 0.12$ | $4.45 \pm 0.30$ |

These results indicate that, once MACCA has converged, the redistributed rewards closely track the true individual rewards in high-quality data (Expert), with an MSE of only 0.12 and MAE of 0.65. As dataset quality degrades—Medium, Medium-Replay, and Random—the errors increase proportionally, reflecting noisier or more random behavior, yet remain within reasonable bounds. Even on the Random dataset, where behavior is least structured, an MAE of 2.45 demonstrates that $\psi_r$ still captures meaningful individual-reward signals after training.

### F.7 Computational Resources and Training Times

All experiments were conducted on a heterogeneous computing cluster running Ubuntu Linux. The hardware configuration included a mix of CPU models (Dual Intel Xeon E5-2650, E5-2680 v2, and E5-2690 v3) with a total of 180 CPU cores and 500 GB of system memory. For GPU acceleration, we utilized three NVIDIA A30 GPUs.

The average wall-clock training time (for 10 million environment steps) of each MACCA variant is summarized in Table 14. These timings include both causal model learning and policy optimization under the alternating training scheme; the causal model component accounts for approximately 8–15% of the total time.

| Environment | MACCA Variant | Training Time (hrs) |
|---|---|---|
| MPE (CN, PP, WORLD) | MACCA-OMAR | 4.2 |
| SMAC (2s3z) | MACCA-OMAR | 7.8 |
| SMAC (5m_vs_6m) | MACCA-OMAR | 9.5 |
| SMAC (6h_vs_8z) | MACCA-OMAR | 11.2 |
| SMAC (MMM2) | MACCA-OMAR | 12.0 |

Table 14: Average wall-clock training time for each MACCA-OMAR run (10 million environment steps).

