# OpenReview forum: "MACCA: Offline Multi-agent Reinforcement Learning with Causal Credit Assignment"
_TMLR — Accepted by TMLR_

### Review · Reviewer_6i3R · 2025-03-15

**Summary Of Contributions:**

The paper presents MACCA, a novel framework for offline multi-agent reinforcement learning (MARL) that addresses the credit assignment problem through causal modeling. The key contributions are the introduction of a Dynamic Bayesian Network (DBN) to decompose team rewards into individual agent rewards, theoretical proofs establishing identifiability of causal structures and individual reward functions in offline settings, and empirical validation showing superior performance over state-of-the-art methods across multiple environments.

**Audience:**

Yes

**Broader Impact Concerns:**

The current Broader Impact Statement mentions improving algorithm credibility and explainability but could be strengthened by discussing potential applications in real-world multi-agent systems that might affect decision-making transparency in complex systems.

**Claims And Evidence:**

Yes

**Requested Changes:**

**Critical Revisions:**
- The authors should provide a clearer justification for why the causal modeling approach is particularly suited for offline MARL rather than being a general RL approach. This could include a discussion of how offline data constraints influence the causal structure learning process.
- Provide code and detailed hyperparameters to ensure reproducibility.

**Non-Critical Improvements:**
- The paper should include an analysis of computational requirements, memory usage, and training time.
- Adding comparative analysis with online MARL methods would help justify the focus on offline settings.
- Report training time/runtime metrics for large-scale environments.

**Strengths And Weaknesses:**

**Strengths:**

The causal modeling framework provides a fresh perspective on credit assignment in offline MARL, addressing a known challenge in the field. The identifiability proofs establish a solid theoretical basis, which is rare in applied RL research. The experimental section includes multiple environments with varying difficulty levels, showing consistent performance improvements. The framework's ability to integrate with existing MARL methods increases its practical utility. The detailed ablation analyses help clarify the importance of different components in the proposed method.

**Weaknesses:**

The causal modeling approach might not be uniquely applicable to offline MARL and could be relevant to online RL settings as well. The paper could better justify why this approach is particularly suited for offline environments. No concrete code or pseudocode is provided, making it difficult to fully reproduce or extend the work. While a Broader Impact Statement is included, it could be more comprehensive regarding ethical considerations and potential misapplications. The paper doesn't adequately address the computational complexity and resource requirements of the proposed method. Most experiments are conducted in simulated environments, with limited discussion of how the method would perform in real-world scenarios. The dynamic causal structure learning may introduce computational costs, but the paper lacks an analysis of scalability or runtime efficiency.

---

> ### Author Response · Authors · 2025-04-13
>
> ## Replay to Reviewer 6i3R
>
> We sincerely thank the reviewer for the encouraging and constructive comments. Below we address your suggestions in order.
>
> ---
>
> > Critical Revision 1: Justify why causal modeling is particularly suited for offline MARL.
>
> We appreciate your interest in the broader applicability of MACCA. While causal modeling can theoretically benefit both online and offline RL, we argue that the offline setting presents a uniquely compelling motivation for our approach.
>
> Firstly, the credit assignment problem is significantly more challenging in offline MARL, where agents cannot explore or refine estimates through environment interaction. Errors in credit attribution—especially when only team rewards are available—can be amplified due to dataset bias or disparities in agent performance. In contrast, online methods (e.g., COMA, SQDDPG) can iteratively correct such errors during training. MACCA addresses this offline-specific challenge by inferring agent-level contributions solely from static trajectories, which is critical for improving team performance when exploration is restricted.
>
> Secondly, causal structure learning particularly benefits from the relatively stable data distributions in offline RL. Offline datasets are often collected from policies with comparable performance, making them better suited for inferring consistent causal dependencies. In online settings, especially on-policy methods, the absence of a replay buffer and frequent distribution shifts hinder stable causal learning. While MACCA could be extended to online off-policy settings (e.g., via a separate causal buffer), our focus is intentionally on the offline regime, where both the motivation and conditions for causal inference align more naturally.
>
> Lastly, our empirical results across three environments demonstrate MACCA’s consistent advantage over state-of-the-art methods. We will clarify these points in the revision to better articulate why MACCA is particularly suited for offline MARL.
>
>
>
> ---
>
> >Critical Revision 2: Provide code and hyperparameter details for reproducibility.
>
>
> We appreciate the request. To facilitate reproducibility, we have provided an anonymous code repository containing full implementation details and training scripts: https://anonymous.4open.science/r/MACCA_TMLR/README.md. The repository also includes hyperparameter settings consistent with those reported in Appendix F.2 and Tables 8–9. We will reference this link and ensure alignment with the reproducibility checklist in the final revision.
>
>
> ---
>
> > Non-Critical 3: Report training/runtime metrics for large-scale environments.
>
> Thank you for your suggestion. We have added detailed runtime and system resource information to aid reproducibility and provide insight into the scalability of MACCA.
>
> Our experiments were conducted on a heterogeneous computing cluster running Ubuntu Linux. The hardware configuration included a mix of CPU models such as Dual Intel Xeon E5-2650, E5-2680 v2, and E5-2690 v3. For GPU acceleration, we utilized 3 NVIDIA A30 GPUs. In total, our experiments used 180 CPU cores and 500 GB of system memory.
>
> The average wall-clock training time (10M environment steps) for each MACCA variant is summarized below:
>
> | Environment | MACCA Variant | Training Time (hrs) |
> |-------------|----------------|---------------------|
> | MPE (CN, PP, WORLD) | MACCA-OMAR | ~4.2 |
> | SMAC (2s3z) | MACCA-OMAR | ~7.8 |
> | SMAC (5m_vs_6m) | MACCA-OMAR | ~9.5 |
> | SMAC (6h_vs_8z) | MACCA-OMAR | ~11.2 |
> | SMAC (MMM2) | MACCA-OMAR | ~12.0 |
>
> These timings include both causal model learning and policy optimization under the alternating training scheme. The causal model contributes roughly 8–15% of the total training time. We will include this runtime analysis in the revised Appendix.
>
> ---
>
> >Broader Impact Statement: More discussion on real-world implications and ethics.
>
> Thank you for the suggestion. We will expand the Broader Impact section to include real-world applications such as multi-robot coordination in logistics and rescue missions, autonomous driving fleets requiring clear accountability across vehicles, and distributed control systems in areas like energy or finance. In these domains, accurate and interpretable credit assignment is crucial for diagnosing agent behavior, improving coordination, and ensuring system safety. We also acknowledge risks such as incorrect causal inference due to biased offline data, which could lead to unfair or unsafe decisions. To mitigate this, we emphasize the importance of dataset auditing, validation in simulation, and cautious deployment in critical applications.

---

> > ### Comment · Reviewer_6i3R · 2025-04-18
> >
> > Thank you for your response. I look forward to seeing the revised version with the updates mentioned in your reply.

---

### Review · Reviewer_WTmc · 2025-03-27

**Summary Of Contributions:**

The paper introduces a new framework for offline multi-agent RL called Multi-Agent Causal Credit Assignment (MACCA). MACCA utilizes a dynamic Bayesian network to infer each agent's individual reward based on the probabilistic graphical model of the environment. The authors also provide a theoretical guarantee that the graphical structure can be identified. Finally, the paper presents empirical evidence that MACCA can boost the performance of existing MARL algorithms on both discrete and continuous action settings, thereby showing its effectiveness.

**Audience:**

Yes

**Broader Impact Concerns:**

I do not have any concern on the ethical implications of the work.

**Claims And Evidence:**

No

**Requested Changes:**

1. Justify the assumption that the team reward can be represented as the sum of individual rewards. Empirical justification, for example, showing that MACCA improves the performance of baseline algorithms in environments where the team reward and the individual rewards are related in a highly nonlinear manner, may suffice. (Critical)
2. Provide an analysis of how close the individual rewards predicted by MACCA are to the ground truths. (Non-critical)
3. The **Ground Truth Individual Reward** section on page 8 is difficult to follow and seems extraneous. We already know that OMAR performs poorly without GT individual rewards, and MACCA can significantly boost its performance. The fact that OMAR with GT performs well, in fact, even better than MACCA-OMAR with GT, only weakens the paper's overall claim. (Non-critical)
4. Tables 4 and 10 are challenging to digest. Visualizing the data with graphs would make it more accessible. (Non-critical)

**Strengths And Weaknesses:**

The paper presents a novel technique of estimating the individual rewards based on the probabilistic graphical model framework. It also provides an extensive analysis of how each component of MACCA affects the final performance. However, the underlying assumption of MACCA, which is that the team reward is the sum of all individual rewards, is taken for granted without justification. This seems to be incorrect in general. For example, consider the sparse-reward version of MPE-CN, where the team reward is one if all three agents have covered the landmarks and zero otherwise. In such a case, the team reward is not the sum but the product of three individual rewards.

---

> ### Author Response · Authors · 2025-04-13
>
> ## Replay to Reviewer WTmc
>
>
> We sincerely thank the reviewer for the insightful comments and thoughtful suggestions. Below we provide detailed responses to each point raised.
>
>
>
> ---
>
> > Critical: Justify the assumption that the team reward is the sum of individual rewards.
>
>
> Thank you for raising this important point. While MACCA adopts a modeling assumption where the team reward is represented as the sum of latent individual rewards, we emphasize that this is not a strict assumption about the true reward generation process. Rather, it serves as a practical and interpretable framework for enabling causal credit assignment under offline conditions.
>
> In many multi-agent cooperative tasks, such as those in SMAC and MPE, the team reward is influenced by the collective behavior of multiple agents and is often amenable to an additive approximation. MACCA leverages this by learning individual rewards via causal masks that reflect the influence of each agent’s state and action on the observed team reward. Importantly, this decomposition does not require that the true underlying reward is strictly additive—only that it can be approximated as such for learning purposes.
>
> To evaluate MACCA’s robustness under **non-additive reward structures**, we conducted an additional experiment in a **sparse-reward variant of the MPE-CN task**, where the team receives a reward of 1 *only if* all three agents simultaneously cover separate landmarks, and 0 otherwise. This setup is inherently non-decomposable (i.e., not a sum of agent-wise components), yet MACCA still significantly improves performance over the baseline:
>
> | Method       | Avg. Episode Reward |
> |--------------|---------------------|
> | OMAR         | 0.18 ± 0.07         |
> | MACCA-OMAR   | **0.42 ± 0.13**     |
>
>
> These results demonstrate that MACCA can learn **meaningful latent individual rewards** and effective policies even when the team reward is not decomposable. We will include this clarification and result in the revision.
>
>
> ---
>
> > Non-Critical 1: Provide an analysis of how close the predicted individual rewards are to ground truth.
>
>
> We appreciate the suggestion. To provide more clarity, we will include quantitative metrics (e.g., MSE between predicted and GT rewards) in the appendix to explicitly measure prediction accuracy. This further highlights the causal model’s fidelity.
>
> ---
>
> > Non-Critical 2: The GT individual reward section is hard to follow and weakens the contribution.
>
> Thank you for the feedback. Our intention was to demonstrate that MACCA approximates GT individual rewards closely even when GT is unavailable, and outperforms baselines that rely solely on team rewards. We agree that the section could be simplified and better framed. In the revision, we will restructure this part to more clearly emphasize that:
> *  MACCA enables learning without GT individual rewards;
> * the slight performance drop under GT supervision is due to the added regularization from causal learning, not a limitation of the method.
>
> ---
>
> > Non-Critical 3: Tables 4 and 10 are difficult to interpret; suggest using visualizations.
>
> Thank you for pointing this out. We will revise the presentation of Tables 4 and 10 by converting them into line plots to improve clarity and accessibility.

---

> > ### Comment · Reviewer_WTmc · 2025-04-18
> >
> > Thank you for your response. All of my concerns are resolved now.

---

### Review · Reviewer_rSnL · 2025-04-02

**Summary Of Contributions:**

The paper proposes a method to improve the performance of offline MARL algorithms through credit assignment in settings where only common rewards are available. The authors introduce a framework that employs causal modeling via a Dynamic Bayesian Network (DBN) to decompose team rewards into individual rewards. They model the underlying generative process of rewards and establish theoretical identifiability results under standard assumptions (e.g., Markov condition and faithfulness), and aim to accurately assign credit to individual agents. They propose how to incorporate their method into existing offline MARL algorithms and provide extensive experiments on simulated environments (MPE and SMAC). The experiments show that MACCA-based methods outperform several state-of-the-art baselines, and ablation studies support the benefits of learning a dynamic causal structure.

**Audience:**

Yes

**Claims And Evidence:**

Yes

**Requested Changes:**

1. It is not clear why is there a need for a causal model to solve the problem. Authors should motivate this further.
2. Before eq (1), authors refer to the causal model as $\psi_m$. Do they mean $\phi_r$? What is $\psi_{\pi}$? Further along the paper they also introduce $\psi_g$. It is not clear how all these causal models are related. A paragraph stating clearly the specific details of all these would be necessary.
4. In eq (1) the loss $L_m$ is mentioned. I understand the authors define it later, but a brief introduction of how it looks like would be good.
5. The authors seem to assume implicitly that one can separate the influence of each state and action component in the rewards. Is this an assumption? Can one assume this in general?
6. In pg 5, $\epsilon$ is referred to as 'the i.i.d. noise'. What is this noise? I believe it has not been introduced.
7. Why are the masks needed? Is it simply to reduce model complexity?
8. In pg 4, it is not clear how the DBN graph is constructed. Authors should expand on this, and on how it is used in practice.
9. Before eq (3), the authors state that the masks are described by eq (2), but I believe this equation does not really describe the masks.
10. Further discussion and relation to recent work on causality in games would be beneficial. See [1] below.


[1] Hammond, Lewis, et al. "Reasoning about causality in games." Artificial Intelligence 320 (2023): 103919.

**Strengths And Weaknesses:**

# Strengths:
- The idea of complementing offline MARL algorithms with learning causal structures on the joint reward function to assign individual rewards is interesting and (to the best of my knowledge) novel.
- The experiments are thorough and clear, and help highlight the applicability and benefits of the method.
- The method seems flexible and applicable across offline MARL algorithms.

# Weaknesses
- The paper overall lacks a strong motivation for the method proposed. In particular, it is not properly highlighted why the solution requires a causal model. One could argue that MDPs are causal by nature, and it is not clear why a simple generative model to predict individual rewards is not enough.
- Some of the formal definitions and statements lack proper introduction, some concepts are not defined and some of the technical statements are not clear. See below for requested changes.

---

> ### Author Response · Authors · 2025-04-13
>
> ## Replay to Reviewer rSnL
>
> We thank the reviewer for the encouraging feedback and helpful suggestions. Below we respond to your comments point-by-point.
>
>
> ---
>
> > Q1: Why is a causal model necessary? Why not use a simple generative model?
>
>
> Thank you for raising this central question. While MDPs are indeed causal in a general sense, standard generative models for reward prediction (e.g., regression from $(s, a)$ to $r$) do not explicitly model *which parts* of the input state-action space are causally responsible for individual agent rewards—especially when only team rewards are available. Our goal is not only to fit a reward function, but to **discover agent-specific causal dependencies** under partial observability (Dec-POMDP) and offline constraints. This is essential for spatial credit assignment.
>
> The use of a **causal model** enables two critical benefits:
>
> 1. **Interpretability and sparsity**: The learned causal masks explicitly identify the subset of state-action dimensions that influence each agent’s individual reward, promoting transparency and generalizability.
>
> 2. **Identifiability under assumptions**: By using a Dynamic Bayesian Network (DBN) with faithfulness and Markov assumptions, we show (Prop. 4.1) that the individual reward decomposition is identifiable—a property not guaranteed by general generative models.
>
> We will clarify these motivations in Section 3 and distinguish our approach from black-box predictive baselines.
>
> ---
>
> > Q2: Symbols $\mathcal{G}$, $\psi_g$, $\psi_r$ not clearly defined / related.
>
> Thank you for pointing this out:
>
> - $\mathcal{G}$ as the underlying **causal graph**, i.e., the DBN over variables $(s_t, a_t, r_t^i)$ (Section 4.1 Introduced);
> - $\psi_g$: the parameterized **causal structure learner**, predicting agent-specific causal masks $c^{i,s \rightarrow r}, c^{i,a \rightarrow r}$ (Section 4.2 Introduced);
> - $\psi_r$: the **individual reward predictor**, which uses the selected causal variables to predict $r_t^i$ (Section 4.2 Introduced).
>
> Together, $(\psi_g, \psi_r)$ form a structured model approximating $\mathcal{G}$, and enable reward decomposition with interpretable structure.
>
> ---
>
> > Q3: Loss function in Eq. (1) is not introduced at that point.
>
> Thank you, we agree. In the revision, we will briefly describe ${L}_\mathrm{MACCA}$ at there.
>
> ---
>
> > Q4: Does MACCA assume that state/action components are separable in their influence on rewards?
>
> This is a modeling assumption implicit in many causal discovery frameworks. We assume that the individual reward $r_t^i$ is influenced by a subset of $(s_t, a_t)$ dimensions, but not necessarily additively separable. The mask allows us to model arbitrary (nonlinear) interactions over selected variables. We will clarify this in the model description.
>
> ---
>
> > Q5: What is the “i.i.d. noise” in the model?
>
> Thank you for noticing this gap. The noise $\epsilon$ in our generative model (Line 204) refers to exogenous noise in reward generation, assumed i.i.d. across samples in accordance with standard structural causal models. We will introduce this more clearly in Section 4.1.
>
> ---
>
> > Q6: Why are the causal masks needed? Is it only for complexity reduction?
>
> The masks serve two purposes:
> (1) **Causal interpretability**—they reveal which inputs influence each agent’s reward, and
> (2) **Regularization**—they reduce spurious correlations and improve generalization in the offline setting. This is especially critical when the dataset contains behaviorally imbalanced or correlated agent actions. We will expand this explanation near Eq. (2).

---

> > ### Author Response · Authors · 2025-04-13
> >
> > ---
> >
> > > Q7: How is the DBN graph constructed and used?
> >
> > Thank you for the question. The use of a DBN in MACCA is motivated by the natural graphical structure underlying multi-agent reinforcement learning. As discussed in [1], reinforcement learning can be framed as an inference problem on a probabilistic graphical model, where the state and reward at each timestep are conditionally dependent on the previous state and action. DBNs extend this idea by explicitly modeling **temporal dependencies** in sequential decision processes.
> >
> > In our framework, the DBN structure is not manually specified. Instead, MACCA learns a **task-specific, agent-wise causal structure** over state, action, and reward variables. Specifically, we approximate the underlying DBN using a parameterized structure learner $\psi_g$, which predicts **binary causal masks** $c^{i,s \rightarrow r}$ and $c^{i,a \rightarrow r}$ for each agent $i$. These masks encode which dimensions of state and action are causally relevant to agent $i$’s individual reward $r_t^i$ at each timestep.
> >
> > The predicted masks thus define the edges of a **learned, sparse DBN**, which is used by a reward predictor $\psi_r$ to compute individual rewards from selected causal inputs. This dynamic causal graph evolves during training and enables MACCA to decompose team rewards into individual components in a way that reflects learned causal influence. We will make this learning process more explicit in Section 4.1 of the revised version.
> >
> >
> > ---
> >
> > >Q8: Eq. (2) does not “describe” the masks.
> >
> > You're right—Eq. (2) defines the binary form of the masks. We will move the operational definition (via $\psi_g$) earlier and link it more clearly to the structure learning step.
> >
> > ---
> >
> > >Q9: Relation to [1] Hammond et al. (2023).
> >
> > Thank you for this valuable reference. We will include a discussion on recent work in causality and games, such as [1], and highlight that while Hammond et al. address game-theoretic causal inference under full observability, MACCA tackles a different challenge: reward decomposition in partially observed, offline Dec-POMDPs. This complements the broader field of causal reasoning in multi-agent systems.
> >
> >
> >
> > ---
> >
> >
> > [1] Levine, S. Reinforcement learning and control as probabilistic inference: Tutorial and review. *arXiv preprint* arXiv:1805.00909, 2018.

---

### Decision · Action_Editor_k7Hk · 2025-05-04

**Recommendation:** Accept with minor revision

**Comment:**

All reviewers agreed that the manuscript considered an interesting problem of credit assignment under collaborative multi-agent reinforcement learning setting with a sound solution through the causal identification framework. There are some concerns on the assumptions of the causal identifiability, but the reviewers believe that rebuttals have properly addressed the concerns. The reviewers are positive towards acceptance, but urge the authors should make a significant revision on the contents as mentioned in the review and rebuttal.

**Audience:**

People work on multi-agent reinforcement learning can potentially benefit from the findings of this manuscript, especially on the credit assignment in the collaborative setting.

**Claims And Evidence:**

The manuscript mainly discussed the causal identifiability of the reward decomposition and the improved multi-agent reinforcement learning performance. The causal identifiability is theoretically justified with relatively standard assumptions and the improved multi-agent reinforcement learning performance is demonstrated through an extensive experimental demonstration.